# Age-Dependent Alterations in Platelet Mitochondrial Respiration

**DOI:** 10.3390/biomedicines11061564

**Published:** 2023-05-28

**Authors:** Zdeněk Fišar, Jana Hroudová, Martina Zvěřová, Roman Jirák, Jiří Raboch, Eva Kitzlerová

**Affiliations:** Department of Psychiatry, First Faculty of Medicine, Charles University and General University Hospital in Prague, Ke Karlovu 11, 120 00 Prague, Czech Republic; hroudova.jana@gmail.com (J.H.); zverova.martina@vfn.cz (M.Z.); roman.jirak@vfn.cz (R.J.); jiri.raboch@lf1.cuni.cz (J.R.); eva.kitzlerova@vfn.cz (E.K.)

**Keywords:** aging, platelet, mitochondria, respiratory chain complex, mitochondrial respiration, cognitive decline, neurodegenerative disease, neuroinflammation, neuroplasticity, oxidative stress

## Abstract

Mitochondrial dysfunction is an important cellular hallmark of aging and neurodegeneration. Platelets are a useful model to study the systemic manifestations of mitochondrial dysfunction. To evaluate the age dependence of mitochondrial parameters, citrate synthase activity, respiratory chain complex activity, and oxygen consumption kinetics were assessed. The effect of cognitive impairment was examined by comparing the age dependence of mitochondrial parameters in healthy individuals and those with neuropsychiatric disease. The study found a significant negative slope of age-dependence for both the activity of individual mitochondrial enzymes (citrate synthase and complex II) and parameters of mitochondrial respiration in intact platelets (routine respiration, maximum capacity of electron transport system, and respiratory rate after complex I inhibition). However, there was no significant difference in the age-related changes of mitochondrial parameters between individuals with and without cognitive impairment. These findings highlight the potential of measuring mitochondrial respiration in intact platelets as a means to assess age-related mitochondrial dysfunction. The results indicate that drugs and interventions targeting mitochondrial respiration may have the potential to slow down or eliminate certain aging and neurodegenerative processes. Mitochondrial respiration in platelets holds promise as a biomarker of aging, irrespective of the degree of cognitive impairment.

## 1. Introduction

The biology of aging is associated with metabolic and oxidative stress, inflammation, and DNA mutations, in which a number of cellular mechanisms are involved [1]. Twelve molecular cellular hallmarks for aging are proposed and discussed, which are interconnected among each other, and which include mitochondrial dysfunction [2]. Mitochondria are organelles that play a key role in bioenergetics and the maintenance and regulation of all brain functions, including neuroinflammation, neuroplasticity, oxidative stress, and apoptosis [3]. In aging, there is evidence of decreased mitochondrial function, increased oxidative stress, increased mitochondrial DNA (mtDNA) mutations, and miRNA dysregulation [4]. Due to the high energy demands of brain cells, mitochondrial dysfunction is associated not only with aging [5], but also with neurodegenerative diseases [6,7]. Age-related neurodegeneration involves complex interplay and synergy of various genetic and environmental factors [8], including mitochondrial impairment as a common motif in the pathophysiology of neuropsychiatric diseases [9]. Mitochondrial dysfunction is associated with decreasing neuroplasticity and weakening functional resilience [10], neuroinflammation [11], oxidative stress and apoptosis [12], excitotoxicity, neurotoxicity of protein agglomerates, and deficiencies in mitochondrial proteostasis and the protease-mediated quality control system [13].

Impaired mitochondrial function can be caused by genetic mutations and epigenetic modifications, environmental stressors, cellular senescence, and disturbed mitophagy. Mitochondrial dysfunction leading to reduced ATP production, increased ROS production, activation of the intrinsic apoptotic pathway, and impaired calcium buffering ultimately result in brain cell death, neurodegeneration, and cognitive decline [6,14].

The main current strategies for preventing or reversing processes associated with aging and neurodegeneration are aimed at mitochondrial quality control mechanisms to increase mitochondrial functions and include regulation of the OXPHOS system, generation of ATP through the electron transport system (ETS), reduction of oxidative stress with antioxidants, inhibition of apoptosis, autophagy enhancement, and stimulation of mitochondrial biogenesis by metabolic modulators, drugs, diet (caloric restriction), and exercise [15,16].

The mitochondrial hypothesis is based on the key role of mitochondria in the production of ATP through oxidative phosphorylation (OXPHOS) [17], the production of reactive oxygen species (ROS), the buffering of free calcium in the cytosol, the release of pro-apoptotic factors and the initiation of the intrinsic pathway of apoptosis, and the production of heat. According to the free radical theory of aging [18,19,20], aging and age-related diseases are associated with the generation of ROS, mainly from mitochondria, and subsequent damage to cellular proteins, lipids, and nucleic acids. Oxidative stress is accepted as a key modulator of the biological processes of aging and neurodegeneration [21], but useful endogenous mechanisms that can be initiated by ROS must also be taken into account in therapeutic interventions on cellular redox processes [22]. Attention is therefore paid to the role of other manifestations of mitochondrial dysfunction, such as inflammation, mtDNA, mitophagy, and retrograde signaling from the mitochondria to the nucleus [23,24,25,26]. 

The mitochondrial hypothesis of aging suggests that mitochondrial dysfunction over time leads to a decrease in cellular energy production, increased oxidative stress, calcium dysregulation, accumulation of mutations in mitochondrial DNA (mtDNA), apoptosis, alterations in mitochondrial dynamics (fusion and fission), accumulation of cellular waste products, disturbed mitophagy, and changes in cellular metabolism [27,28,29,30]. This, in turn, may be the biological basis for the development of age-related diseases, including neurodegenerative diseases, and the possibility of their treatment [31,32,33,34].

Mitochondrial dysfunctions are implicated in the pathophysiology of neurodegenerative diseases, such as Alzheimer’s disease (AD), Parkinson’s disease, and Huntington’s disease [6,31,35]. Common features in their pathogenesis are mitochondrial dysfunction, progressive neurodegeneration, and cognitive decline leading to dementia. The main risk factors for the onset and development of AD are age, genetics and epigenetics, environmental factors, amyloidosis, tauopathy, and mitochondrial dysfunction. Neurodegenerative mechanisms in AD include early changes in mitochondria-associated endoplasmic reticulum membranes [36]. In the integrative hypothesis of Alzheimer’s disease, interlinking and mutual synergistic connections between amyloid beta pathology, tau pathology, mitochondrial dysfunction, dysfunction of neurotransmitter systems, and disturbed neuroplasticity are assumed [37].

Mitochondrial dysfunction is also thought to contribute to the pathophysiology of psychiatric diseases such as major depressive disorder (MDD) [38] and bipolar disorder (BD) [39,40,41,42] through several pathways, including oxidative stress, neuroinflammation, genetics, and disturbed neuroplasticity. Affective disorders are a common comorbidity in neurodegenerative diseases [43]. Cognitive deficit is a principal component in MDD, especially in late-life depression [44,45]. Cognitive impairment in BD patients is associated with metabolic factors, gene polymorphisms, brain structural and functional changes, and neuroinflammation [46]. Emotions can improve or impair cognitive performance [47], and it is important to understand the functional interplay between the central and autonomic nervous systems and to elucidate how emotions are integrated into executive functions in neuropsychiatric diseases [48,49]. This is crucial for regulating physiological processes and maintaining homeostasis, with the prefrontal cortex playing a key role [50,51].

There are several assays to measure mitochondrial dysfunction in aging and neurodegeneration [7,52,53,54], and they use different biological models. It was proved that platelets are a suitable biological model for research on mitochondrial dysfunction [55,56,57]. Platelets are small blood components (nonnucleated cells) in mammals derived from the megakaryocytes. Average life span of circulating platelets is 8 to 9 days [58]; therefore, they reflect current systemic changes. Platelets contribute to hemostasis, innate immunity, and inflammatory response [59]. It was shown that platelet bioenergetics reflect muscle energetics and platelet mitochondrial function is altered in older adults [60]. Moreover, platelets are considered peripheral elements reflecting a variety of brain functions and neurochemical changes, including those leading to neurodegeneration [61,62,63]. Platelets can be easily separated from blood as platelet rich plasma (PRP) and mitochondrial function can be measured under physiological conditions [57]. Since platelets share many properties with brain cells [64], they appear to be a suitable biological model for monitoring processes of brain aging and neurodegeneration.

Age dependence of mitochondrial function in various organs and brain regions in rats has been described [65,66,67]; the results did not confirm the concept of a general pattern of age-dependent mitochondrial dysfunction. No such data are available in humans; measurements in humans are usually performed using skeletal muscle biopsies, fibroblasts, and circulating blood cells [68].

Due to intracellular homeostatic mechanisms, it is appropriate to monitor changes in both the activity of individual mitochondrial proteins and enzyme complexes, as well as complex mitochondrial functions characterizing a real physiological state. Frequently used methods for measuring mitochondrial function include measuring the rate of oxygen consumption, ATP production, hydrogen peroxide production, free calcium buffering in the cytosol, and the release of proapoptotic factors. In this study, the measurement of the activity of citrate synthase (CS), complexes I, II, III, and IV of the respiratory chain, and the parameters of mitochondrial respiration in the platelets of healthy individuals and individuals with neuropsychiatric disease is used to evaluate the age-dependent changes of mitochondrial dysfunction.

## 2. Materials and Methods

Commonly used mitochondrial function assays include oxygen consumption (complex I- and II-linked respiration, respiratory control ratio, uncoupling) [57,69], ATP production [70], hydrogen peroxide production [71,72], membrane potential [73,74], mitochondrial permeability transition, swelling, and calcium retention capacity [75,76], monoamine oxidase [77], release of pro-apoptotic factors (by ELISA and chromatography), membrane fluidity [78], mtDNA mutations [79], mitochondrial morphology and dynamics [80], mitochondrial biogenesis, and mitophagy [81].

Age-dependent changes in mitochondrial function were measured in people of different ages using spectrophotometric and high-resolution respirometry methods. Activity of CS, complex I, II, III, and IV and mitochondrial respiratory rate was measured in blood platelets of healthy subjects (CONTROL) and patients with Alzheimer’s disease (AD), vascular dementia (VD), major depressive disorder (MDD), or bipolar disorder (BAD). For the analysis of the age dependence of measured mitochondrial parameters, data from control subjects included in our earlier analyses and publications on platelet mitochondrial parameters were used. Data from patients with AD, VD, MDD, or BAD were used to evaluate changes in age dependence of platelet mitochondrial parameters in people with cognitive impairment. The material and methods used have been published previously, so they are presented here only very briefly with the corresponding references.

### 2.1. Subjects and Their Clinical Evaluation

The subjects and their clinical evaluation have been described previously for AD patients [82,83,84,85], VD patients [83], MDD patients [86,87], and BAD patients [87,88]. Patients with AD, VD, MDD, or BAD were diagnosed and recruited from the Department of Psychiatry of the First Faculty of Medicine, Charles University and General University Hospital in Prague, Czech Republic. In each individual study, the number of participants was determined to ensure adequate statistical power required to detect differences between groups of healthy subjects and neuropsychiatric patients.

Patients with probable AD or probable VD over 60 years of age were recruited. Criteria to diagnose AD and VD included the International Classification of Diseases, Tenth Edition (ICD-10), NINDS-AIREN VD criteria [89], the NINCDS-ADRDA Alzheimer’s criteria [90,91], and the Hachinski Ischemic Score [92]. Depressive symptoms in AD patients were assessed by the Geriatric Depression Scale (GDS) [93]. Disease severity was assessed by the Addenbrooke’s Cognitive Examination-Revised [94] inclusive of the Mini-Mental State Examination (MMSE) questionnaire. Other causes of dementia than AD or VD were excluded. Patients with a BAD diagnosis were clinically evaluated using diagnostic scales and questionnaires, including the Young Mania Rating Scale (YMRS), Clinical Global Impression—Severity Scale (CGI-01), and Brief Psychiatric Rating Scale (BPRS). The severity of depression in MDD patients was evaluated using the Hamilton Depressive Rating Scale, 21-item (HDRS-21) and the Clinical Global Impression—severity scale (CGI-01).

The controls included healthy volunteers, who underwent a psychiatric examination that was equivalent to that of neuropsychiatric patients, and they were without cognitive decline. The participants did not take mitochondria-targeting compounds.

Cognitive decline (decline in memory, attention, language, and executive function) is a common feature of aging and neuropsychiatric diseases. The Mini-Mental State Examination (MMSE) total score has been used to assess the progression of cognitive impairment as follows: no cognitive impairment with MMSE 24–30; mild cognitive impairment with MMSE 19–23; moderate cognitive impairment with MMSE 10–18; and severe cognitive impairment with MMSE ≤ 9 [95].

### 2.2. Chemicals and Solutions and Measurement Methods

The chemicals, solutions, and measurement methods are described in our earlier publications. Activities of CS and respiratory chain complexes were measured spectrophotometrically [84,96,97,98]; mitochondrial respiration in platelets was measured by high-resolution respirometry using the Oxygraph-2k (Oroboros Instruments Corp, Innsbruck, Austria) [57,99]. All chemicals were purchased from Sigma-Aldrich Co. (St. Louis, MO, USA)

Studies were conducted in accordance with the Code of Ethics of the World Medical Association (Declaration of Helsinki), and the study protocols were approved by the Ethical Review Board of the First Faculty of Medicine, Charles University and General University Hospital in Prague. Written informed consent was obtained from participants.

### 2.3. Data Analysis

Statistical analyses were performed using the STATISTICA data analysis software system (TIBCO Software Inc., Palo Alto, CA, USA). The relation between the measured platelet parameters and age was determined using the Pearson product-moment correlation coefficient. Simple regression was used to quantify the age dependence of all measured mitochondrial parameters.

DatLab software (Oroboros Instruments Corp, Innsbruck, Austria) was used for respirometry data acquisition and analysis. Oxygen consumption rates were normalized for platelet concentration (pmol O_2_ per sec per 10^6^ platelets). Activities of mitochondrial complexes I, II, III, and IV were normalized for CS activity.

## 3. Results

The age dependence of the mitochondrial parameters measured in platelets isolated from peripheral blood was evaluated. The following parameters were measured in platelets of healthy subjects (CONTROL) and patients with AD, VD, MDD, or BAD: (1) CS activity; (2) complex I activity normalized for CS (CI/CS); (3) complex II activity normalized for CS (CII/CS); (4) complex III activity normalized for CS (CIII/CS); (5) complex IV activity normalized for CS (CIV/CS); (6) oxygen consumption rate in various respiratory states. Mitochondrial respiration in intact platelets was determined as (i) routine (basal) respiration (ROUT_i); (ii) oligomycin-induced respiration independent of ADP phosphorylation (LEAK_i); (iii) maximum capacity of electron transport system (ETSC_i); (iv) respiratory rate after complex I inhibition (ROT_i); and (v) using a derived parameter called respiration reserve capacity (RES_i) and calculated as ‘ETSC_i-ROUT_i’.

A total of 637 blood samples from people aged 20 to 94 were included in the evaluation, including 162 samples of control subjects aged 20 to 80 years without serious somatic and neuropsychiatric diseases, 196 samples of AD patients aged 50 to 94 years, 120 samples of subjects with MDD aged 20 to 74 years, 127 subjects with BAD aged 20 to 73 years, and 32 people with VD aged 65 to 85 years. The age dependence of the mitochondrial parameters was calculated using correlation and regression analysis.

Correlation coefficients (Pearson r) between age and the mitochondrial parameters are summarized in Table 1. In the group of healthy controls, there is a significant negative correlation between age and CS, CII/CS, ROUT_i, ETSC_I, ROT_i, and RES_i; there is a significant positive correlation between age and CIV/CS.

The age dependence of the measured parameters (ROUT_i, LEAL_i, ETSC_i, ROT_i, and RES_i) was quantified using the slopes of regression lines. For control group members, but not for AD, MDD, and BAD patients, a significant dependence on age was found for CS, CII/CS, and CIV/CS (Table 2, Figure 1), ROUT_i, ETSC_i, ROT_i, and RES_i (Table 3, Table 4, Table 5, Table 6 and Table 7). The greatest statistically significant decrease with age is shown in the mitochondrial respiratory parameter ETSC_i (Figure 2). Information was added to the results that the slope (linear regression result) for the age dependence of platelet count normalized for CS is not significantly different from zero.

To assess the effect of depression on the age dependence of mitochondrial respiration, regression slopes were calculated for the subgroup of AD patients with depression (AD + DEP) and without depression (AD − DEP). Depression in AD was diagnosed for a Geriatric Depression Score > 6 [93]. A significant negative slope was found for the age dependence of ROUT_i and ROT_i in AD + DEP but not in AD − DEP (Table 3 and Table 6).

The effect of cognitive impairment on the age dependence of mitochondrial respiration was determined by dividing all included controls and neuropsychiatric patients into three groups according to their MMSE score: (1) no cognitive impairment with MMSE > 23 (*N* = 328), (2) mild cognitive impairment with MMSE in the interval <19,23> (*N* = 37), and (3) moderate or severe cognitive impairment with MMSE < 19 (*N* = 52). A significant negative slope was found for ROUT_i, ETSC_i, ROT_i, and RES_i in persons without significant cognitive impairment but not in persons with cognitive impairment (Table 3, Table 4, Table 5, Table 6 and Table 7, Figure 3).

## 4. Discussion

The age dependence of the mitochondrial parameters was assessed for citrate synthase activity, activity of respiratory chain complexes, and kinetics of mitochondrial oxygen consumption in platelets. The effect of cognitive impairment on the age dependence of the mitochondrial parameters was evaluated by comparing healthy individuals and individuals with neuropsychiatric disease (AD, VD, MDD, and BAD). Because it is not known whether the progression of mitochondrial dysfunction is different in neuropsychiatric diseases, we separately evaluated data for controls, AD, VD, MDD, and BAD, as well as pooled data. The progression of changes in mitochondrial respiration was evaluated using simple regression. We chose the significance and magnitude of the slope of the regression line as a criterion for evaluating the effect of age on the value of the mitochondrial parameter.

The significant correlation between age and measured mitochondrial parameters was negative in subjects without neuropsychiatric disease, except for complex IV activity normalized for CS (Table 1), whereas unnormalized complex IV activity did not change significantly with age. Because (i) CS activity normalized for platelet count does not show a significant correlation with age and (ii) CS and complex I activities are associated with mitochondrial content, while complex IV activity is associated with OXPHOS capacity [100], our data indicate that the reduction of mitochondrial content with age is due to a reduced number of platelets, rather than a reduced content of mitochondria in platelets. Regression analysis quantified the results by correlations (Table 2, Table 3, Table 4, Table 5, Table 6 and Table 7). This indicates that the activity of individual mitochondrial enzymes and enzyme complexes involved in OXPHOS system, as well as the complex parameters of mitochondrial respiration, decrease with age. The increase in complex IV activity of the respiratory chain with age may be part of a compensatory mechanism to maintain cellular homeostasis. Non-significant correlations between age and measured mitochondrial parameters in persons with neuropsychiatric diseases (Table 1) indicate a small influence of these diseases on mitochondrial dysfunction associated with aging.

Our results indicate that the values of the mitochondrial respiration parameters ROUT_i, ETSC_i, LEAK_i, and RES_i decrease significantly with age, while the time course is not significantly different in controls and in neuropsychiatric diseases such as AD, VD, BAD, and MDD.

Depressive disorder is a frequent comorbidity in AD [101,102] and significantly reduces mitochondrial respiration in intact platelets. Therefore, age dependence was assessed separately in the subgroup of AD patients with and without depression. It can be speculated that the significant negative slope for ROUT_i, ETSC_i, ROT_i in AD patients with depression (Table 5 and Table 6) is associated with comorbid depressive disorder.

The impairment of mitochondrial respiration in intact platelets was previously described in AD but was not associated with disease progression [82]. A significant negative slope for the age dependence of the respiratory parameters ROUT_i, ETSC_i, ROT_i, and RES_i in persons without cognitive impairment and statistically insignificant changes in slopes in persons with cognitive impairment (Table 3, Table 4, Table 5, Table 6 and Table 7) indicate that mitochondrial dysfunction (manifested by changes in mitochondrial oxygen consumption in intact platelets) is probably not a measure of cognitive impairment in AD. A certain limitation for this conclusion is the inclusion of a relatively low number of persons with severe cognitive impairment in the analysis.

The age dependence of mitochondrial respiration on cognitive impairment supports the hypothesis that progressive neurodegeneration in AD is associated with specific neurotoxicity of amyloid beta oligomers and tau protein, rather than direct consequences of mitochondrial dysfunction. However, age-related neuropsychiatric diseases can be initiated and regulated by mitochondrial dysfunction [37].

The results support a possible role of mitochondrial dysfunction in the regulation of aging processes. The progression of mitochondrial dysfunction (determined from measurements of mitochondrial respiration in intact platelets) does not appear to be significantly altered. Assuming that mitochondrial respiration in intact platelets reflects mitochondrial changes in the brain, the reduction of mitochondrial respiration does not appear to be the primary cause of neurodegeneration associated with aging but is only part of complex processes associated with impaired brain neuroplasticity.

In summary, a significant decrease with age was found for both the activity of mitochondrial enzymes (CS and CII) and the parameters of mitochondrial respiration measured in intact platelets (ROUT_i, ETSC_i, ROT_i, and RES_i). Compared to the large interindividual differences in these parameters, the rate of their reduction is not too large with age, and it can be expected that in healthy aging mitochondrial dysfunction can be regulated to a certain extent by lifestyle, primarily by exercise and diet [103,104,105].

Age-related neuropsychiatric diseases, such as AD, are associated with a decrease in mitochondrial respiration in intact platelets [82,84], but no significant change in age dependence of respiratory parameters was observed in platelets from AD patients compared to healthy controls (Figure 3). This supports the notion that the direct contribution of mitochondrial dysfunction to neurodegeneration is similar in aging and in AD, and that AD progression is driven more by amyloid beta and tau pathology. Mitochondrial dysfunction remains a promising candidate as an initial trigger and/or synergistic promoter of this specific pathophysiology [37]. For pathology of neurodegenerative diseases that have already begun, the therapeutic strategy should therefore include, in addition to dietary restriction and exercise (discussed under healthy aging), targeted pharmacological regulation of the causes of the development of specific disease pathology. To confirm the role of mitochondrial dysfunction in neurodegeneration associated with healthy aging and age-related neuropsychiatric diseases, repeated measurements of appropriate mitochondrial variables must be performed in the same persons over long periods of time. Suitable variables could be parameters of mitochondrial respiration in intact platelets.

### 4.1. Study Limitations

A limitation of mitochondrial dysfunction research in aging and neurodegeneration is the difficult identification of specific mitochondrial mechanisms and interactions with other cellular causes of aging and neurodegeneration. Mitochondrial biomarkers of aging and neurodegeneration are therefore still being sought. Some limitations in the study of age-dependent mitochondrial dysfunction include lack of standardized methods for measuring mitochondrial dysfunction. The development of standardized methods for measuring mitochondrial respiration is currently being intensively addressed. Most measurements have been conducted using animal models and cell lines. However, the use of intact platelets from peripheral blood [57] used in this study may solve the limited availability of human tissue samples to measure mitochondrial dysfunction. Although this is a good model for monitoring changes in a number of biochemical parameters in the brain that manifest themselves systemically, we must be aware that this is a model of brain changes. For comprehensive assessment of mitochondrial dysfunction during aging, a combination of commonly used methods should be used, including mitochondrial respiration, ROS production, mtDNA mutations, mitochondrial membrane potential, mitochondrial morphology and dynamics, mitochondrial biogenesis and mitophagy (see Introduction).

In this study, mitochondrial respiration was normalized for platelet concentration, and activities of mitochondrial complexes were normalized for CS activity. CS activity can be used as a marker for mitochondrial dysfunction, and a decline in CS activity has been linked to a decrease in the number of mitochondria and changes in mitochondrial morphology [100]. CS activity in the muscles may decrease with aging [106,107]. Thus, CS activity may be used to normalize other mitochondrial parameters, such as the activity of respiratory chain complexes [108] and mitochondrial respiration. However, using CS activity to normalize mitochondrial data means additional measurements and a possible source of error. Due to the good correlation between CS activity and platelet count in PRP, the normalization of mitochondrial respiration for platelet count appears to be suitable for measurements in intact platelets [82]. In the case of evaluating the overall activity of the OXPHOS system (measured by mitochondrial respiration), we consider normalization to the platelet count in the sample to be more accurate and simpler.

### 4.2. Clinical Significance

There are several treatment strategies to prevent or to slow down cognitive decline due to age or neurodegenerative disease. This is not only about medication (such as cholinesterase inhibitors, NMDA receptor antagonists, or targeting glutamatergic, noradrenergic, and endocannabinoid systems) [109], but also about exercise, diet, cognitive training, and social engagement.

Mitochondrial dysfunctions measured as disruption of mitochondrial respiration in intact platelets are common features of aging and neurodegenerative diseases, such as AD. Our study showed that the progression of mitochondrial respiration reduction with age does not appear to be significantly affected by neurodegenerative disease. The strategy of improving mitochondrial function in aging and neurodegeneration may include regulation of the OXPHOS system at the level of stimulation of mitochondrial biogenesis, activity of individual mitochondrial enzymes of the citrate cycle and respiratory chain complexes, and changing the availability of substrates of the electron transport system.

The research on age-dependent mitochondrial dysfunction has both theoretical implications and translational applications. Theoretical implications of age-dependent mitochondrial dysfunction research include understanding the mechanisms of aging and neurodegeneration. Translational applications of this research include finding new biomarkers for the diagnosis of age-related diseases and for finding molecular targets of new drugs effective in preventing or mitigating cognitive decline and neurodegeneration. Understanding the mechanism of onset and progression of mitochondrial dysfunction in aging and neurodegeneration can lead to the development of new therapies that target specific mitochondrial processes and are able to slow down the onset of age-related diseases.

### 4.3. Research Perspectives

Both aging and mitochondrial dysfunction are complex processes regulated by a variety of factors, including lifestyle and genetic, epigenetic, and internal and external environmental factors. The perspective in the research on mitochondrial dysfunction during aging and neurodegeneration therefore consists in a multidisciplinary approach combining genetic, biochemical, and physiological measurements with clinical evaluation of neurodegeneration and cognitive impairment. In addition, it is necessary to consider other biological processes that contribute to the process of aging and neurodegeneration, including neuroinflammation, apoptosis, and neurotoxicity. More accurate and detailed assessment of mitochondrial dysfunction in aging and neurodegeneration will be provided by advances in technologies for measuring mitochondrial function, further development of animal models of age-related diseases, development of methods for measuring mitochondrial biomarkers in peripheral blood, and application of omics technologies.

The regulation of mitochondrial function, neuroplasticity, and neurotransmission are means of prospective prevention and treatment of neurodegenerative diseases. However, careful monitoring of side and adverse effects of potential drugs is necessary, as stimulation of mitochondrial functions can also lead to negative neuroplasticity. Finally, there are large interindividual differences in mitochondrial dysfunction during aging and neurodegeneration, and treatment/regulation should therefore be based on a personalized approach.

## 5. Conclusions

Study of mitochondrial dysfunction during aging and neurodegeneration is an important area of research with the perspective of potential therapeutic use of new knowledge. The findings showed the potential of measuring mitochondrial respiration in intact platelets to assess age-related mitochondrial dysfunction and support the role of mitochondrial dysfunction in the aging process. Mitochondrial respiration in platelets may serve as a biomarker for aging and cognitive decline and it can be expected that interventions that improve mitochondrial function may prevent or slow cognitive decline in aging and age-related diseases. Mitochondrial respiration in platelets appears to be a potential biomarker of aging, regardless of the degree of cognitive impairment.

## Figures and Tables

**Figure 1 biomedicines-11-01564-f001:**
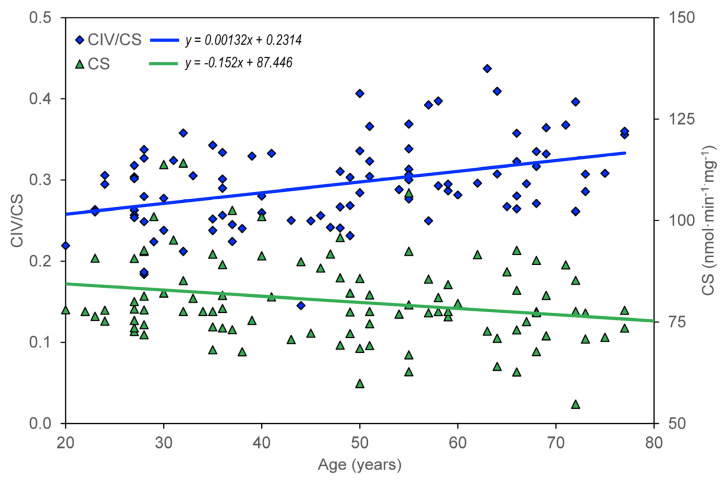
Age dependence of citrate synthase activity (CS) and complex IV activity normalized for CS (CIV/CS) in healthy controls.

**Figure 2 biomedicines-11-01564-f002:**
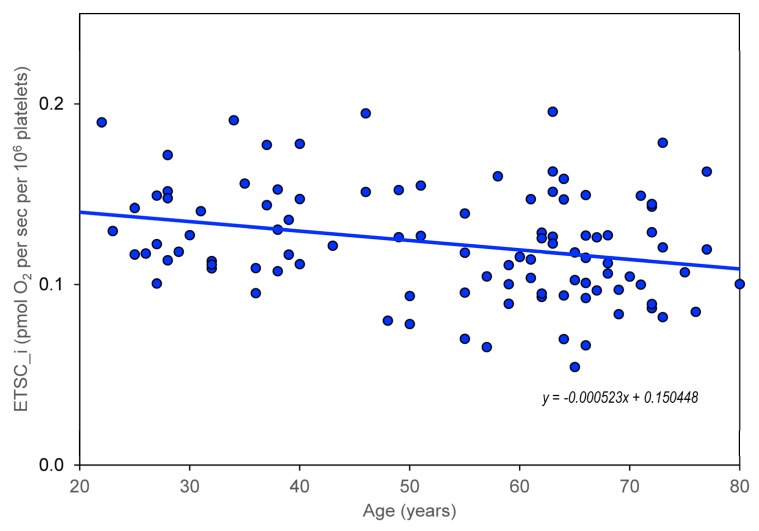
Age dependence of maximal capacity of electron transport system in intact platelets (ETSC_i) in healthy controls.

**Figure 3 biomedicines-11-01564-f003:**
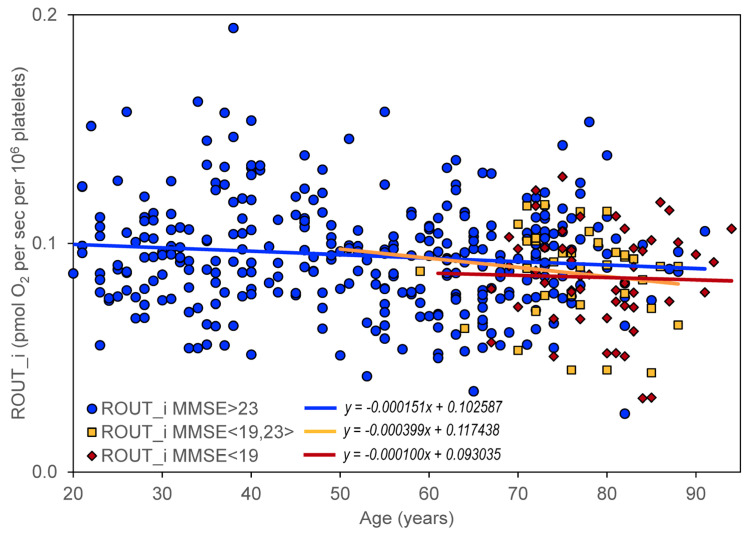
Age dependence of routine respiration state in intact platelets (ROUT_i) in subjects without cognitive impairment (MMSE > 23), with mild cognitive impairment (MMSE in the interval <19,23>), and with moderate or severe cognitive impairment (MMSE < 19).

**Table 1 biomedicines-11-01564-t001:** Correlation coefficients (Pearson r) between age and platelet mitochondrial variables.

Parameter	CONTROL	AD	MDD	BAD	VD	All
CS	**−0.2317**	−0.0104	−0.1735	−0.0149	--	−0.0817
	*N* = 100	*N* = 86	*N* = 111	*N* = 98	*N* = 0	*N* = 395
	*p* = 0.020	*p* = 0.924	*p* = 0.069	*p* = 0.884	*p* = −	*p* = 0.105
CI/CS	0.0435	0.1380	0.1283	−0.0321	−−	−0.0728
	*N* = 93	*N* = 81	*N* = 107	*N* = 97	*N* = 0	*N* = 378
	*p* = 0.679	*p* = 0.219	*p* = 0.188	*p* = 0.755	*p* = −	*p* = 0.158
CII/CS	**−0.2529**	−0.0625	−0.0211	0.0603	−−	0.0135
	*N* = 93	*N* = 82	*N* = 107	*N* = 96	*N* = 0	*N* = 378
	*p* = 0.014	*p* = 0.577	*p* = 0.829	*p* = 0.559	*p* = −	*p* = 0.794
CIII/CS	0.2142	−0.2139	−0.0533	−−	−−	**0.3205**
	*N* = 31	*N* = 22	*N* = 15	*N* = 0	*N* = 0	*N* = 68
	*p* = 0.247	*p* = 0.339	*p* = 0.850	*p* = −	*p* = −	*p* = 0.008
CIV/CS	**0.4072**	−0.1393	0.0791	0.0023	−−	**0.1360**
	*N* = 93	*N* = 83	*N* = 107	*N* = 98	*N* = 0	*N* = 381
	*p* < 0.001	*p* = 0.209	*p* = 0.418	*p* = 0.982	*p* = −	*p* = 0.008
ROUT_i	**−0.2048**	−0.0326	−0.0860	−0.0943	−0.1567	**−0.1617**
	*N* = 101	*N* = 132	*N* = 42	*N* = 113	*N* = 29	*N* = 417
	*p* = 0.040	*p* = 0.710	*p* = 0.588	*p* = 0.320	*p* = 0.417	*p* = 0.001
LEAK_i	0.0919	−0.0224	**−0.3418**	0.0091	−0.0806	**0.1150**
	*N* = 101	*N* = 132	*N* = 42	*N* = 113	*N* = 29	*N* = 417
	*p* = 0.361	*p* = 0.799	*p* = 0.027	*p* = 0.923	*p* = 0.678	*p* = 0.019
ETSC_i	**−0.2812**	0.0360	0.1013	−0.0602	−0.1964	**−0.2663**
	*N* = 101	*N* = 132	*N* = 42	*N* = 113	*N* = 29	*N* = 417
	*p* = 0.004	*p* = 0.682	*p* = 0.523	*p* = 0.527	*p* = 0.307	*p* < 0.001
ROT_i	**−0.3688**	0.0810	0.2249	0.0851	−0.0810	−0.0881
	*N* = 101	*N* = 132	*N* = 42	*N* = 113	*N* = 29	*N* = 417
	*p* = 0.000	*p* = 0.356	*p* = 0.152	*p* = 0.370	*p* = 0.676	*p* = 0.072
RES_i	**−0.2137**	0.0967	0.2001	0.0151	−0.1811	**−0.2187**
	*N* = 101	*N* = 132	*N* = 42	*N* = 113	*N* = 29	*N* = 417
	*p* = 0.032	*p* = 0.270	*p* = 0.204	*p* = 0.874	*p* = 0.347	*p* = 0.000

Mitochondrial parameters were measured in platelets of healthy subjects (CONTROL) and patients with Alzheimer’s disease (AD), vascular dementia (VD), major depressive disorder (MDD), or bipolar disorder (BAD). Statistically significant correlation coefficients are in bold. CS, citrate synthase activity (nmol·min^−1^·mg^−1^); CI/CS, complex I activity normalized for CS; CII/CS, complex II activity normalized for CS; CIII/CS, complex III activity normalized for CS; CIV/CS, complex IV activity normalized for CS; ROUT_i, routine (basal) respiration; LEAK_i, oligomycin-induced respiration independent of ADP phosphorylation; ETSC_i, maximum capacity of electron transport system; ROT_i, respiratory rate after complex I inhibition by rotenone; and RES_i, respiration reserve capacity calculated as ‘ETSC_i-ROUT_i’ (all respiration parameters in pmol O_2_·10^−6^ platelets·sec^−1^).

**Table 2 biomedicines-11-01564-t002:** Age dependence of platelet concentration and mitochondrial enzyme activities.

Sample	Parameter	Slope	95% CI	*p*	Age range	*N*
CONTROL	PRP	−794	(−2400, 813)	0.331	20–80	162
	CS	*** −0.152**	(−0.280, −0.024)	**0.020**	20–77	100
	CI/CS	0.00055	(−0.00209, 0.00320)	0.679	20–77	93
	CII/CS	*** −0.00067**	(−0.00120, −0.00014)	**0.014**	20–77	93
	CIII/CS	0.00153	(−0.00112, 0.00418)	0.247	23–77	31
	CIV/CS	***** 0.00132**	(0.00070, 0.00194)	**0.000**	20–77	93
AD	PRP	−2009	(−4306, 288)	0.086	50–94	196
	CS	−0.020	(−0.426, 0.387)	0.924	56–91	86
	CI/CS	0.00482	(−0.00293, 0.01258)	0.219	56–91	81
	CII/CS	−0.00036	(−0.00166, 0.00093)	0.577	56–91	82
	CIII/CS	−0.00375	(−0.01175, 0.00424)	0.339	56–88	22
	CIV/CS	−0.00174	(−0.00448, 0.00100)	0.209	56–91	83
MDD	PRP	−728	(−2531, 1075)	0.426	20–74	120
	CS	−0.217	(−0.452, 0.017)	0.069	20–74	111
	CI/CS	0.00178	(−0.00088, 0.00445)	0.188	20–74	107
	CII/CS	−0.00006	(−0.00060, 0.00048)	0.829	20–74	107
	CIII/CS	−0.00038	(−0.00470, 0.00393)	0.850	29–74	15
	CIV/CS	0.00044	(−0.00063, 0.00152)	0.418	20–74	107
BAD	PRP	−686	(−2808, 1437)	0.524	20–73	127
	CS	−0.021	(−0.305, 0.263)	0.884	21–66	98
	CI/CS	−0.00180	(−0.01321, 0.00962)	0.755	21–66	97
	CII/CS	0.00044	(−0.00105, 0.00193)	0.559	21–66	96
	CIII/CS	ND	ND	ND	ND	0
	CIV/CS	0.00002	(−0.00179, 0.00183)	0.982	21–66	98
VD	PRP	5408	(−12379, 1561)	0.123	65–85	32
All	PRP	***** −1606**	(−2205, −1006)	**<0.001**	20–94	637
	CS	−0.073	(−0.162, 0.015)	0.105	20–91	395
	CI/CS	−0.00172	(−0.00412, 0.00067)	0.158	20–91	378
	CII/CS	0.00004	(−0.00027, 0.00036)	0.794	20–91	378
	CIII/CS	**** 0.00233**	(0.00064, 0.00402)	**0.008**	23–88	68
	CIV/CS	**** 0.00064**	(0.00017, 0.00111)	**0.008**	20–91	381

Analyzed with simple regression. Statistically significant differences compared with controls is presented as * *p* < 0.05, ** *p* < 0.01, and *** *p* < 0.001. Statistically significant values are in bold. 95% CI, 95% confidence interval; PRP, platelet rich plasma. The meaning of the abbreviations is the same as in Table 1.

**Table 3 biomedicines-11-01564-t003:** Age dependence of mitochondrial respiration parameter ROUT_i.

Sample	Slope (95% CI)		*p*	Age Range	Valid *N*
CONTROL	*** −0.309**	(−0.604, −0.014)	**0.040**	22–80	101
AD	−0.087	(−0.547, 0.374)	0.710	50–94	132
MDD	−0.013	(−0.308, 0.281)	0.927	20–74	42
BAD	−0.175	(−0.522, 0.172)	0.320	21–66	113
VD	−0.617	(−2.153, 0.919)	0.417	65–85	29
All	***** −0.212**	(−0.328, −0.095)	**0.000**	20–94	417
AD + DEP	*** −1.067**	(−2.129, −0.005)	**0.049**	60–85	37
AD − DEP	0.023	(−0.474, 0.520)	0.927	50–94	95
MMSE > 23	*** −0.151**	(−0.295, −0.006)	**0.041**	20–91	328
MMSE <19,23>	−0.399	(−1.461, 0.663)	0.451	59–88	37
MMSE < 19	−0.100	(−1.049, 0.850)	0.834	67–94	52

Analyzed with simple regression. Statistically significant differences compared with controls is presented as * *p* < 0.05 and *** *p* < 0.001. Statistically significant values are in bold. DEP, depression; MMSE, Mini-Mental State Examination. The meaning of the other abbreviations is the same as in Table 1 and Table 2. Slope is given in fmol O_2_·10^−6^ platelets·sec^−1^·year^−1^.

**Table 4 biomedicines-11-01564-t004:** Age dependence of mitochondrial respiration parameter LEAK_i.

Sample	Slope (95% CI)		*p*	Age Range	Valid *N*
CONTROL	0.038	(−0.044, 0.121)	0.361	22–80	101
AD	−0.013	(−0.118, 0.091)	0.799	50–94	132
MDD	*** −0.117**	(−0.219, −0.014)	**0.027**	20–74	42
BAD	0.004	(−0.071, 0.078)	0.923	21–66	113
VD	−0.076	(−0.448, 0.296)	0.678	65–85	29
All	*** 0.033**	(0.005, 0.060)	**0.019**	20–94	417
AD + DEP	−0.009	(−0.221, 0.203)	0.933	60–85	37
AD − DEP	−0.013	(−0.138, 0.113)	0.841	50–94	95
MMSE > 23	0.033	(−0.001, 0.067)	0.058	20–91	328
MMSE <19,23>	−0.047	(−0.298, 0.205)	0.707	59–88	37
MMSE < 19	0.008	(−0.214, 0.231)	0.941	67–94	52

Analyzed with simple regression. Statistically significant differences compared with controls is presented as * *p* < 0.05. Statistically significant values are in bold. The meaning of the abbreviations is the same as in Table 1, Table 2 and Table 3.

**Table 5 biomedicines-11-01564-t005:** Age dependence of mitochondrial respiration parameter ETSC_i.

Sample	Slope (95% CI)		*p*	Age Range	Valid *N*
CONTROL	**** −0.523**	(−0.879, −0.167)	**0.004**	22–80	101
AD	0.106	(−0.404, 0.616)	0.682	50–94	132
MDD	0.163	(−0.347, 0.673)	0.523	20–74	42
BAD	−0.155	(−0.640, 0.329)	0.527	21–66	113
VD	−1.057	(−3.139, 1.026)	0.307	65–85	29
All	***** −0.412**	(−0.555, −0.268)	**0.000**	20–94	417
AD + DEP	−0.502	(−1.639, 0.636)	0.377	60–85	37
AD − DEP	0.041	(−0.503, 0.585)	0.881	50–94	95
MMSE > 23	**** −0.300**	(−0.483, −0.116)	**0.001**	20–91	328
MMSE <19,23>	−0.213	(−1.302, 0.876)	0.694	59–88	37
MMSE < 19	0.276	(−0.674, 1.226)	0.562	67–94	52

Analyzed with simple regression. Statistically significant differences compared with controls is presented as ** *p* < 0.01 and *** *p* < 0.001. Statistically significant values are in bold. The meaning of the abbreviations is the same as in Table 1, Table 2 and Table 3.

**Table 6 biomedicines-11-01564-t006:** Age dependence of mitochondrial respiration parameter ROT_i.

Sample	Slope (95% CI)		*p*	Age Range	Valid *N*
CONTROL	***** −0.154**	(−0.231, −0.076)	**0.000**	22–80	101
AD	0.059	(−0.067, 0.185)	0.356	50–94	132
MDD	0.080	(−0.031, 0.191)	0.152	20–74	42
BAD	0.026	(−0.032, 0.085)	0.370	21–66	113
VD	−0.082	(−0.481, 0.317)	0.676	65–85	29
All	−0.028	(−0.058, 0.003)	0.072	20–94	417
AD + DEP	**** 0.346**	(0.108, 0.584)	**0.006**	60–85	37
AD − DEP	−0.026	(−0.176, 0.123)	0.728	50–94	95
MMSE > 23	*** −0.042**	(−0.078, −0.006)	**0.023**	20–91	328
MMSE <19,23>	−0.145	(−0.487, 0.197)	0.394	59–88	37
MMSE < 19	0.032	(−0.260, 0.324)	0.827	67–94	52

Analyzed with simple regression. Statistically significant differences compared with controls is presented as * *p* < 0.05, ** *p* < 0.01, and *** *p* < 0.001. Statistically significant values are in bold. The meaning of the abbreviations is the same as in Table 1, Table 2 and Table 3.

**Table 7 biomedicines-11-01564-t007:** Age dependence of mitochondrial respiration parameter RES_i.

Sample	Slope (95% CI)		*p*	Age Range	Valid *N*
CONTROL	*** −0.214**	(−0.409, −0.019)	**0.032**	22–80	101
AD	0.193	(−0.151, 0.536)	0.270	50–94	132
MDD	0.197	(−0.127, 0.522)	0.226	20–74	42
BAD	0.020	(−0.224, 0.263)	0.874	21–66	113
VD	−0.440	(−1.382, 0.503)	0.347	65–85	29
All	***** −0.201**	(−0.285, −0.117)	**0.000**	20–94	417
AD + DEP	0.565	(−0.192, 1.322)	0.139	60–85	37
AD − DEP	0.018	(−0.374, 0.410)	0.928	50–94	95
MMSE > 23	**** −0.148**	(−0.249, −0.046)	**0.004**	20–91	328
MMSE <19,23>	0.186	(−0.636, 1.008)	0.649	59–88	37
MMSE < 19	0.376	(−0.363, 1.115)	0.312	67–94	52

Analyzed with simple regression. Statistically significant differences compared with controls is presented as * *p* < 0.05, ** *p* < 0.01, and *** *p* < 0.001. Statistically significant values are in bold. The meaning of the abbreviations is the same as in Table 1, Table 2 and Table 3.

## Data Availability

The datasets generated and analyzed during the current study are not publicly available due to the fact that individual privacy could be compromised, but anonymized data are available from the corresponding author on reasonable request.

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
