# Peer review of "Age-Dependent Alterations in Platelet Mitochondrial Respiration"

_biomedicines, 2023, doi:10.3390/biomedicines11061564_

Round 1

Reviewer 1 Report

Fisar and colleagues in the present article entitled ‘Age dependence of mitochondrial respiration in platelets in relation to cognitive decline’, investigated the parameters of mitochondrial respiration in the platelets of healthy individuals and individuals with neuropsychiatric disease is used to evaluate the age dependence of mitochondrial dysfunction.

In general, I think the idea of this article is really interesting and the authors’ fascinating observations on this timely topic may be of interest to the readers of Biomedicines. However, some comments, as well as some crucial evidence that should be included to support the author’s argumentation, needed to be addressed to improve the quality of the manuscript, its adequacy, and its readability prior to the publication in the present form. My overall judgment is to publish this paper after the authors have carefully considered my suggestions below, in particular reshaping parts of the ‘Introduction’ and ‘Methods’ sections by adding more evidence.

Please consider the following comments:

A graphical abstract that will visually summarize the main findings of the manuscript is highly recommended.

Abstract: In my opinion, Authors should consider rephrasing this section. According to the Journal’s guidelines, the Abstract should contain most of the following kinds of information in brief form. Please, consider giving a more synthetic overview of the paper's key points: I would suggest rephrasing the results and conclusion to make them clear for readers to understand. Also, please provide an abstract which contains no more than 200 words.

In general, I recommend authors to use more references to back their claims, especially in the Introduction of this article, which I believe is lacking. Thus, I recommend the authors to attempt to expand the topic of their article, as the bibliography is too concise. Nevertheless, I believe that less than 60/70 articles are too low for a research article. Therefore, I suggest the authors to focus their efforts on researching relevant literature: in my opinion, adding more citations will help to provide better and more accurate background to this study. 

Introduction: This section is well-written and nicely presented, with a good balance of descriptive text and information about dysfunctions in neurobiological mechanisms of aging. Nevertheless, I believe that more information about how mitochondrial damage and oxidative stress have been greatly implicated in the progression of aging, along with the pathogenesis of age-related neurodegenerative diseases (NDs), would provide a better and more accurate background. Thus, I would suggest to make such effort to provide a brief overview of the pertinent published data on how mitochondrial dysfunction has a crucial role in the pathophysiology of age-related neurodegeneration, resulting in decreased effectiveness of cellular mechanisms, initiation of inflammatory pathways, excitotoxicity, protein agglomeration and apoptosis (https://doi.org/10.1016/j.neubiorev.2023.105163; DOI: 10.3390/ijms24065926).

Subjects: In my opinion, data about participants and information about clinical assessment for patients’ selection are not adequately explained. For this reason, I would ask the authors to specify inclusion criteria for patients involved in this study, like severity of disorder. Also, could the authors specify how did they estimate the exact number of participants and provide more information about the diagnostic tests used for clinical evaluation?

Discussion: In this final section, authors described the results of their study and their argumentation and captured the state of the art well; however, I would have liked to see some views on a way forward. I believe that the authors should make an effort, trying to explain the theoretical implication as well as the translational application of this paper, to adequately convey what they believe is the take-home message of their study. In this regard, I believe that it would be necessary to discuss theoretical and methodological avenues in need of refinement, as well as suggestions of a path forward in the understanding of the role of alteration in signaling cascades of apoptosis in mitochondrial dysfunction and the therapeutic strategies (both natural and synthetic drugs) targeting these mitochondrial apoptotic pathways and oxidative stress that holds great promise.

I think the ‘Conclusions’ paragraph would benefit from some thoughtful as well as in-depth considerations by the authors, because as it stands, it lists down all the main findings of the research, without really stressing the theoretical significance of the study. Authors should make an effort, trying to explain the theoretical implication as well as the translational application of their research.

In according to the previous comment, I would ask the authors to include a proper and defined ‘Limitations and future directions’ section before the end of the manuscript, in which authors can describe in detail and report all the technical issues brought to the surface.

Tables and Figures: According to the Journal’s guidelines, please provide a short explanatory caption for the table within the text.

References: Authors should consider revising the bibliography, as there are several incorrect citations. Indeed, according to the Journal’s guidelines, they should provide the abbreviated journal name in italics, the year of publication in bold, the volume number in italics for all the references.

I hope that, after these careful revisions, this paper can meet the Journal’s high standards for publication. 

I am available for a new round of revision of this article. 

Best regards,

Reviewer

Author Response

A list of changes to each point raised by reviewers

Journal name: Biomedicines

Manuscript ID: biomedicines-2387170

Title: Age-dependent changes in mitochondrial respiration in platelets

Authors: ZdenÄ›k Fišar, Jana Hroudová, Martina Zvěřová, Roman Jirák, JiÅ™í Raboch, and Eva Kitzlerová

We are submitting a revised original research article, "Age-dependent changes in mitochondrial respiration in platelets" (Manuscript ID: biomedicines-2387170) for publication in Biomedicines (special Issue “10th Anniversary of Biomedicines-Neurological Diseases and Mental Illness”). We would like to thank the Academic Editor and Reviewers for their valuable comments and suggestions for improving the manuscript. We edited the manuscript according to all comments from the Editor and Reviewers.

Reviewer 1:

Please consider the following comments:

  • A graphical abstract that will visually summarize the main findings of the manuscript is highly recommended.

             Response:

  • Graphical abstract has been added.

  • Abstract: In my opinion, Authors should consider rephrasing this section. According to the Journal’s guidelines, the Abstract should contain most of the following kinds of information in brief form. Please, consider giving a more synthetic overview of the paper's key points: I would suggest rephrasing the results and conclusion to make them clear for readers to understand. Also, please provide an abstract which contains no more than 200 words.

             Response:

  • The abstract has been modified.

  • In general, I recommend authors to use more references to back their claims, especially in the Introduction of this article, which I believe is lacking. Thus, I recommend the authors to attempt to expand the topic of their article, as the bibliography is too concise. Nevertheless, I believe that less than 60/70 articles are too low for a research article. Therefore, I suggest the authors to focus their efforts on researching relevant literature: in my opinion, adding more citations will help to provide better and more accurate background to this study. 

             Response:

  • More references are cited.

  • Introduction: This section is well-written and nicely presented, with a good balance of descriptive text and information about dysfunctions in neurobiological mechanisms of aging. Nevertheless, I believe that more information about how mitochondrial damage and oxidative stress have been greatly implicated in the progression of aging, along with the pathogenesis of age-related neurodegenerative diseases (NDs), would provide a better and more accurate background. Thus, I would suggest to make such effort to provide a brief overview of the pertinent published data on how mitochondrial dysfunction has a crucial role in the pathophysiology of age-related neurodegeneration, resulting in decreased effectiveness of cellular mechanisms, initiation of inflammatory pathways, excitotoxicity, protein agglomeration and apoptosis (https://doi.org/10.1016/j.neubiorev.2023.105163; DOI: 10.3390/ijms24065926).

             Response:

  • The Introduction has been expanded and restructured. More references are cited.

  • Subjects: In my opinion, data about participants and information about clinical assessment for patients’ selection are not adequately explained. For this reason, I would ask the authors to specify inclusion criteria for patients involved in this study, like severity of disorder. Also, could the authors specify how did they estimate the exact number of participants and provide more information about the diagnostic tests used for clinical evaluation?

             Response:

  • Data about participants and about clinical assessment were briefly supplemented; details are in the cited publications.
  • Discussion: In this final section, authors described the results of their study and their argumentation and captured the state of the art well; however, I would have liked to see some views on a way forward. I believe that the authors should make an effort, trying to explain the theoretical implication as well as the translational application of this paper, to adequately convey what they believe is the take-home message of their study. In this regard, I believe that it would be necessary to discuss theoretical and methodological avenues in need of refinement, as well as suggestions of a path forward in the understanding of the role of alteration in signaling cascades of apoptosis in mitochondrial dysfunction and the therapeutic strategies (both natural and synthetic drugs) targeting these mitochondrial apoptotic pathways and oxidative stress that holds great promise.

             Response:

  • The Discussion has been expanded. Subchapters were added to the Discussion: ‘Study limitations’, ‘Clinical significance’, and ‘Research perspectives’, which explain the theoretical implication as well as the translational application of this paper. Theoretical and methodological avenues are discussed.

  • I think the ‘Conclusions’ paragraph would benefit from some thoughtful as well as in-depth considerations by the authors, because as it stands, it lists down all the main findings of the research, without really stressing the theoretical significance of the study. Authors should make an effort, trying to explain the theoretical implication as well as the translational application of their research.

             Response:

  • Conclusion was edited and reduced to single paragraph.
  • Theoretical implication as well as the translational application of research are now included in the Discussion.

  • In according to the previous comment, I would ask the authors to include a proper and defined ‘Limitations and future directions’ section before the end of the manuscript, in which authors can describe in detail and report all the technical issues brought to the surface.

             Response:

  • Subchapters were added to the Discussion: ‘Study limitations’, ‘Clinical significance’, and ‘Research perspectives’.

  • Tables and Figures: According to the Journal’s guidelines, please provide a short explanatory caption for the table within the text.

             Response:

  • Information was added.

  • References: Authors should consider revising the bibliography, as there are several incorrect citations. Indeed, according to the Journal’s guidelines, they should provide the abbreviated journal name in italics, the year of publication in bold, the volume number in italics for all the references.

             Response:

  • More references are cited.
  • References were reformatted.

Reviewer 2 Report

This is an interesting report describing a careful analysis of of age dependency of respiratory chain function in platelets. 

I have the following comments:

1. In Fig. 3 the different data points are very difficult to distinguish, please use colors. 

2. The authors should discuss the problem that CS decreases with age but COX/CS increases. Does this mean that COX activity stays constant?

3. Respiration activity not normalized to CS decreases. What happens if respiration activity is normalized to CS activity?

4. Finally, it should be discussed if the amount of mitochondria or the activity of ETC enzymes changes with age.

None.

Author Response

A list of changes to each point raised by reviewers

Journal name: Biomedicines

Manuscript ID: biomedicines-2387170

Title: Age-dependent changes in mitochondrial respiration in platelets

Authors: ZdenÄ›k Fišar, Jana Hroudová, Martina Zvěřová, Roman Jirák, JiÅ™í Raboch, and Eva Kitzlerová

We are submitting a revised original research article, "Age-dependent changes in mitochondrial respiration in platelets" (Manuscript ID: biomedicines-2387170) for publication in Biomedicines (special Issue “10th Anniversary of Biomedicines-Neurological Diseases and Mental Illness”). We would like to thank the Academic Editor and Reviewers for their valuable comments and suggestions for improving the manuscript. We edited the manuscript according to all comments from the Editor and Reviewers.

Reviewer 2:

  1. In Fig. 3 the different data points are very difficult to distinguish, please use colors. 

             Response:

  • Figures are now presented in color.

  1. The authors should discuss the problem that CS decreases with age but COX/CS increases. Does this mean that COX activity stays constant?

             Response:

  • The decrease in CS and increase in COX/CS is now discussed.
  • Yes, our data showed almost constant non-normalized COX activity - this is now mentioned in the discussion.
  • Normalization of mitochondrial parameters for CS activity is now discussed.

  1. Respiration activity not normalized to CS decreases. What happens if respiration activity is normalized to CS activity?
  • Information was added to the results that the slope (linear regression result) for age dependence of platelet count normalized for CS is very close to zero.
  • Normalization for platelet count and for CS activity is now discussed.

  1. Finally, it should be discussed if the amount of mitochondria or the activity of ETC enzymes changes with age.

             Response:

  • The change in mitochondrial content in platelets with age is now mentioned in the discussion.

Round 2

Reviewer 1 Report

The authors did an excellent job clarifying all the questions I have raised in my previous round of review. Currently, this paper is a well-written, timely piece of research that described how the parameters of mitochondrial respiration in the platelets of healthy individuals and individuals with neuropsychiatric disease is used to evaluate the age dependence of mitochondrial dysfunction.

Overall, this is a timely and needed work. It is well researched and nicely written, therefore I believe that this paper does not need a further revision, therefore the manuscript meets the Journal’s high standards for publication.

I am always available for other reviews of such interesting and important articles.

Thank You for your work, Reviewer

Author Response

We thank the reviewer for the valuable comments and suggestions in his Review report (Round 1) and we are glad that we were able to comply with all of them.

Reviewer 2 Report

The authors have addressed all of my concerns appropriately.

None.

Author Response

(The authors gave the same response as above.)
